# ArbGaze: Gaze Estimation from Arbitrary-Sized Low-Resolution Images

**DOI:** 10.3390/s22197427

**Published:** 2022-09-30

**Authors:** Hee Gyoon Kim, Ju Yong Chang

**Affiliations:** Department of Electronics and Communications Engineering, Kwangwoon University, Seoul 01897, Korea

**Keywords:** gaze estimation, knowledge distillation, feature adaptation, deep neural network

## Abstract

The goal of gaze estimation is to estimate a gaze vector from an image containing a face or eye(s). Most existing studies use pre-defined fixed-resolution images to estimate the gaze vector. However, images captured from in-the-wild environments may have various resolutions, and variation in resolution can degrade gaze estimation performance. To address this problem, a gaze estimation method from arbitrary-sized low-resolution images is proposed. The basic idea of the proposed method is to combine knowledge distillation and feature adaptation. Knowledge distillation helps the gaze estimator for arbitrary-sized images generate a feature map similar to that from a high-resolution image. Feature adaptation makes creating a feature map adaptive to various resolutions of an input image possible by using a low-resolution image and its scale information together. It is shown that combining these two ideas improves gaze estimation performance substantially in the ablation study. It is also demonstrated that the proposed method can be generalized to other popularly used gaze estimation models through experiments using various backbones.

## 1. Introduction

The human eye gaze tells what a person’s interests are, and it can be used as a medium for non-verbal communication. As a means of communication, the gaze can be used in many areas, such as human–computer interaction [1], human–robot interaction [2], virtual reality [3], augmented reality [4], and autonomous driving [5]. The goal of the gaze estimation task in computer vision is to estimate the gaze information of a subject from an input image that includes the subject’s face or eye(s). The gaze information can be expressed as a target display’s two-dimensional (2D) pixel coordinates to which the subject’s gaze is fixated or as a three-dimensional (3D) gaze vector in the camera coordinate system. The target display varies depending on the environment in which the image is acquired, such as a mobile phone [6], a laptop [7], and a tablet [8].

Existing gaze estimation methods estimate gaze information from images acquired in a controlled environment or from synthetic images [9,10,11,12,13,14,15,16]. Many studies assume that the resolution of the face or eye(s) extracted from the input image is similar. In addition, high-resolution facial images obtained in environments where the distance between the camera and the subject is relatively close (e.g., mobile phones, tablets, and laptops) are available for gaze estimation [6,7,17,18]. However, in an in-the-wild environment where the distance between the camera and the subject is not constant or in an environment where the distance between the subject and the target display is relatively far (e.g., TV), low-resolution images of various sizes can be acquired. These low-resolution input images degrade the performance of the gaze estimation method. Our study addresses this problem in gaze estimation from arbitrary-sized low-resolution images, as shown in Figure 1a. It should be noted that the aspect ratio of the input image (i.e., the width ratio to the height of the image) is assumed to be constant in our study. Addressing the variation of the input image’s aspect ratio is beyond our study’s scope.

To address this problem, super-resolution (SR) methods with an arbitrary scale factor [19,20,21] can be utilized. The basic idea is to reconstruct a high-resolution image from a low-resolution image and feed the resulting image into the gaze estimation model, as shown in Figure 1b. However, the SR method requires much memory and time for training, and noise or artifacts in the reconstructed image can disturb correct gaze estimation. Figure 1c shows another approach in which different gaze estimation models are trained according to each resolution to cover input images of various resolutions. However, this method requires a large number of models corresponding to multiple resolutions.

To efficiently perform gaze estimation from arbitrary-sized low-resolution images, we propose a method (i.e., ArbGaze) that utilizes knowledge distillation [22] and feature adaptation [20] together. In knowledge distillation, the knowledge of a larger network (i.e., teacher) is generally transferred to a small network (i.e., student). On the contrary, the teacher and student have a network structure of almost the same complexity in our study. The difference between the two networks is in their input images: the teacher and student use high-resolution images and arbitrary-sized low-resolution images for gaze estimation, respectively. Therefore, the teacher can learn privileged knowledge that the student cannot obtain from low-resolution images. It makes the intermediate features of the teacher contain more abundant information for gaze estimation than the student. Our basic idea is to distill the teacher’s knowledge into the student, allowing the student to generate features similar to the rich features of the teacher.

However, models with fixed weights do not have sufficient complexity to adapt to input images of arbitrary resolution. Therefore, the proposed student is constructed using the feature adaptation module based on conditional convolution (i.e., CondConv) [23] that has a larger capacity than the general convolution operation. A scale factor can be defined between high-resolution images for the teacher and arbitrary-sized low-resolution images for the student. This scale factor indicates the ratio of the resolution of the high-resolution image and that of the low-resolution image. The larger the scale factor, the greater the difference in size between the two images. The feature adaptation module outputs a new scale-adapted feature map from a feature map obtained from an arbitrary-sized low-resolution image using this scale information. Intuitively, knowledge distillation supervises the student on what features to create, and feature adaptation supervises the student on how to construct those features.

In summary, the research question to be answered in this study is whether efficient gaze estimation is possible from an arbitrary-sized low-resolution image. Our hypothesis is that knowledge distillation and feature adaptation can be utilized together to construct an efficient and effective gaze estimation model. The contributions of this study are as follows.

A novel method is proposed to solve the problem of gaze estimation from low-resolution input images of arbitrary size, which has been rarely addressed previously.The proposed method significantly improves the gaze estimation performance for low-resolution images of various sizes by combining knowledge distillation and feature adaptation.

## 2. Related Works

This section introduces existing gaze estimation studies and problems that may arise in gaze estimation. Knowledge distillation and feature adaptation are utilized in the proposed method for gaze estimation from arbitrary-sized low-resolution input images. Therefore, knowledge distillation and feature adaptation for various scales are briefly reviewed.

### 2.1. Gaze Estimation

Gaze estimation methods can be divided into model-based and appearance-based approaches [6,24,25,26,27]. In the model-based approach, the physical structure of the eye, such as corneal reflex, iris, or pupil movement, is geometrically modeled for gaze estimation [28,29,30]. These methods typically rely on special equipment, such as infrared or RGB-D cameras. On the contrary, the appearance-based approach directly uses input images. It can estimate the gaze information from data acquired through a general camera, such as a webcam [6,7,8,10,11,12,13,14,15,16,17,18,31,32]. Owing to this advantage, the appearance-based approach has dominated gaze estimation studies recently. Such appearance-based methods are now briefly reviewed.

Early gaze estimation methods estimated gaze through various machine learning techniques, such as random forest [9,10], adaptive linear regression [33], and support vector regression [6]. Recently, gaze estimation studies using large-scale datasets [6,7,8,9,10,11,12,14,17,34,35] and deep learning have been actively conducted. Deep learning-based gaze estimation methods can be mainly divided into 2D gaze estimation studies [6,8,36,37,38] and 3D gaze estimation studies [7,10,11,12,13,14,15,18,31]. The goal of 2D gaze estimation is to estimate the 2D coordinates of a pixel on which the gaze is directed on a specific display, such as a cell phone or laptop. In the 3D gaze estimation method, the gaze vector in the camera coordinate system is usually estimated. The gaze vector can be a 3D direction vector [10,11,31] or an angle vector composed of pitch and yaw [12,13,14,15,16,17,18].

In gaze estimation, various issues may arise in addition to the problem we focus on in this paper. Anatomical differences between people can degrade gaze estimation performance. To solve this inter-person problem, ref. [39] proposed a person-specific few-shot learning method using a few calibration samples. In this method, a personalized gaze estimator based on a rotation-aware latent representation of gaze is trained using meta-learning. Ref. [40] addressed the issue of gaze estimation datasets. Building a new gaze estimation dataset is expensive, and the existing datasets are annotated under different environments. Therefore, in [40], an unsupervised representation learning method that does not use ground-truth information was presented. It uses gaze redirection, which generates an output image with the same eye as the input image but with redirected gaze, as an auxiliary task for unsupervised learning. Ref. [41] addressed the difficulty of acquiring 3D gaze information in an in-the-wild environment. One interesting idea to solve this is to utilize the geometric constraint of people looking at each other in the video. In [41], looking-at-each-other labels for in-the-wild images were obtained based on such an idea, and a weakly-supervised training method for gaze estimation based on the labels was proposed. Ref. [42] addressed the domain adaptation problem in gaze estimation. To generalize the gaze estimator to a new domain, ref. [42] proposed a plug-and-play gaze adaptation framework, where outliers of the outputs of pre-trained models in the target domain are considered noisy labels, helping the collaborative learning process.

The issues above are important but not related to the motivation of our study, which is presented as follows. Most appearance-based methods, including the works of literature above, assume that an input image of fixed size is used. Therefore, to use the existing methods, it is necessary to resample the low-resolution images of various sizes into images of a fixed size. Unfortunately, loss of information in the general resampling process can cause deterioration of gaze estimation performance. Our paper addresses this issue and proposes an effective gaze estimation method from an arbitrary-sized low-resolution image.

### 2.2. Knowledge Distillation

A straightforward way to increase the performance of deep learning models is to utilize large-scale datasets with complex networks. However, this requires substantial computation and memory usage. Knowledge distillation is proposed to allow relatively simple models (i.e., student) to achieve similar performance to complex models (i.e., teacher) [22]. As the complex teacher model has high capacity, it can learn richer knowledge from data than the simple student model. This knowledge of the teacher model is transferred to a lightweight student model via a loss function between the two models, which is trained to output intermediate features similar to the teacher model. The teacher–student framework of knowledge distillation is adopted from this perspective for the proposed gaze estimation method. A loss function is applied between the teacher model trained on high-resolution images and the student model trained on arbitrary-sized low-resolution images, causing the student model to output intermediate features similar to the teacher model. These intermediate features contain information that is difficult to learn from low-resolution images, which helps improve the gaze estimation performance of the student model.

### 2.3. Scale- or Resolution-Dependent Feature Adaptation

Consistently addressing low-resolution images of different sizes is a critical issue in various areas of computer vision, including gaze estimation. Ref. [19] proposed a meta upscale module that reconstructs high-resolution images from low-resolution feature maps over multiple scale factors. This module is trained to output weights for image reconstruction using the positional relationship of the corresponding pixels between the low-resolution feature map and the high-resolution image together with the scale factor. For human shape and pose estimation, ref. [43] proposed a resolution-aware network (i.e., RA-net) to handle various resolutions of input images. RA-net adapts the feature map by multiplying the feature map extracted from the input image by a resolution-aware matrix composed of parameters including resolution information. Similar to [19], ref. [20] proposed a feature adaptation block to handle low-resolution images of various sizes in SR. Based on CondConv [23], the feature adaptation block adapts the output feature map of the baseline network to the scale factor by increasing the similarity between intermediate feature maps for low-resolution input images of various sizes. Unlike previous studies addressing SR [19,20] and human shape estimation [43] tasks, feature adaptation is performed for gaze estimation from arbitrary-sized low-resolution images in our study, where the feature adaptation block of [20] is adopted.

## 3. Proposed Method

Our framework consists of a teacher network $N^{T}$ and a student network $N^{S}$, as shown in Figure 2. $N^{T}$ and $N^{S}$ output a 3D gaze vector from fixed-sized ($H^{HR}\times W^{HR}$) high-resolution image $I^{HR}$ and arbitrary-sized ($H^{LR}\times W^{LR}$) low-resolution image $I^{LR}$, respectively. Our framework adopts offline distillation, so $N^{T}$ is trained before $N^{S}$. $I^{LR}$, an input image of $N^{S}$, may have various resolutions. $I^{LR}$ is converted into an image, $I^{BC}$, with the same resolution as the high-resolution image through bicubic interpolation before being processed in the student. Feature adaptation modules ($A_{1}$, …, $A_{N}$) in $N^{S}$ produce intermediate features ($F_{1}^{A}$, …, $F_{N}^{A}$) adaptive to arbitrary resolution using the input scale factor *s*. The scale factor *s* is defined as follows under the assumption that $I^{HR}$ and $I^{LR}$ have the same aspect ratio: (1)$$s≔\frac{H^{HR}}{H^{LR}}=\frac{W^{HR}}{W^{LR}}.$$

In the training of the student $N^{S}$, the teacher $N^{T}$ transfers information that is difficult to learn from low-resolution images to $N^{S}$ through the distillation loss $L_{KD}^{S}$ between the intermediate features of the two networks. Using the feature adaptation module and distillation loss improves the gaze estimation performance of the student $N^{S}$ for arbitrary-sized low-resolution images.

### 3.1. Teacher Network

The teacher network $N^{T}$ outputs a gaze vector ${\hat{g}}^{T}\in R^{3}$ from a high-resolution input image $I^{HR}$. ResNet-18 [44] is used to construct $N^{T}$. As shown in Figure 2, $N^{T}$ consists of one convolutional layer, *N* residual modules, one global average pooling layer, and two fully connected (FC) layers. As a residual module, the building block of [44] is used. The output feature map $F_{n}^{T}$ of the *n*-th residual module $R_{n}^{T}$ is used for the next layer’s input and the distillation loss. The output $F_{N}^{T}$ of the last residual module is passed through the global average pooling layer and two FC layers to produce the estimated gaze vector ${\hat{g}}^{T}$. The loss function $L_{gaze}^{T}$ used to learn $N^{T}$ is defined as the angular error between two gaze vectors, which is as follows:(2)$$L_{gaze}^{T}=\arccos\left(\frac{g\cdot {\hat{g}}^{T}}{{∥g∥}_{2}∥{{\hat{g}}^{T}∥}_{2}}\right),$$
where $g\in R^{3}$ and ${∥\cdot ∥}_{2}$ denote the ground-truth gaze vector corresponding to the input image and 2-norm, respectively.

### 3.2. Student Network

The student network $N^{S}$ outputs a gaze vector ${\hat{g}}^{S}$ from an arbitrary-sized low-resolution image $I^{LR}$ and its scale factor *s*. As shown in Figure 2, $N^{S}$ has a structure in which a bicubic interpolation module and *N* feature adaptation modules ($A_{1}$, …, $A_{N}$) are added to $N^{T}$, where each feature adaptation module is placed after each residual module. First, the bicubic interpolation is applied to an arbitrary-sized low-resolution image $I^{LR}$ to output an image $I^{BC}$ having the same resolution as $I^{HR}$. Then, $I^{BC}$ is fed into the convolutional layer and passed through the residual module and feature adaptation module alternately. $F_{N}^{A}$ computed by the last feature adaptation module is fed into the global average pooling layer, and the gaze vector estimation ${\hat{g}}^{S}$ is generated through two FC layers.

The feature adaptation module produces the feature map adaptive to the scale factor of the input low-resolution image from the feature map computed by the residual module. The feature adaptation module increases the similarity between the intermediate feature maps of the teacher and the student over multiple scale factors and, as a result, allows the student to output teacher-like feature maps over multiple scale factors [20]. The feature adaptation module is composed of a masking block and a scale-conditioned layer, as shown in Figure 3a. The scale-conditioned layer is a 2D convolutional layer based on CondConv [23]. This layer consists of the two FC layers and *K* experts (i.e., $W_{1}$, …, $W_{K}$), as shown in Figure 3b. The result calculated through the linear combination of *K* routing weights (i.e., $\alpha_{1}$, …, $\alpha_{K}$) produced by two FC layers and *K* experts is used for 2D convolution as kernel weights *W*. The masking block consists of four convolutional layers followed by a sigmoid function. The feature map computed by the sigmoid function, composed of values between 0 and 1, is a kind of mask that determines the pixel position of the feature map to which the scale information generated by the scale-conditioned layer is added [20]. From the masking block’s output and the scale-conditioned layer’s output, a feature map including scale information is calculated through element-wise multiplication. Finally, it is added to the input feature map $F_{n}^{S}$ to output a scale-adapted feature $F_{n}^{A}$.

The loss function $L^{S}$ applied to the student network consists of $L_{gaze}^{S}$, which measures the angular error and distillation loss $L_{KD}^{S}$ as follows:(3)$$L^{S}=L_{gaze}^{S}+\lambda L_{KD}^{S},$$
where λ is a hyperparameter that controls the relative strength between the two loss functions. $L_{gaze}^{S}$ is defined similarly to $L_{gaze}^{T}$ in Equation (Equation 2), and the distillation loss $L_{KD}^{S}$ to make the student network generate features similar to the teacher network is defined as follows:(4)$$L_{KD}^{S}=\sum_{n=1}^{N}MSE\left(F_{n}^{T},F_{n}^{A}\right),$$
where MSE(·,·) denotes the mean squared error between two feature maps, and *N* represents the number of feature adaptation modules.

### 3.3. Model Training and Testing

Training the proposed ArbGaze model consists of two steps: (i) training the teacher network using high-resolution images and gaze loss in Equation (Equation 2), and (ii) training the student network using arbitrary-sized low-resolution images and gaze/distillation losses in Equation (Equation 3). When training the student network, the teacher network is used to calculate the distillation loss, but the weights of the teacher network are frozen and not updated. During the test, the teacher network is discarded, and the student network is applied to arbitrary-sized low-resolution test images to output estimated gaze vectors, which are used to evaluate the proposed method.

## 4. Experimental Results

### 4.1. Datasets

The publicly available gaze estimation datasets UTMultiview (UTMV) [10] and MPIIGaze (MPII) [17] are used to evaluate the proposed method. The UTMV dataset provides real and synthetic data. Among them, only real data consisting of 8 head poses and 160 gaze directions for each subject are used. As a result, the dataset contains 64,000 (50×8×160) images obtained from 50 subjects. The MPII dataset is acquired via a laptop in an unconstrained environment and contains 45,000 images from 15 subjects. Both datasets provide images that have undergone the same preprocessing process, and these images are used as high-resolution images (i.e., $I^{HR}$) for training the teacher network. The preprocessing process first rotates a virtual camera to remove the roll rotation from the head pose and to point it toward the center of the eye, then sets the virtual camera at a fixed distance from the subject. A grayscale image of $H^{HR}\times W^{HR}=36\times 60$ resolution for one eye is generated from the virtual camera, and additional histogram equalization is applied to the generated image. Low-resolution images (i.e., $I^{LR}$) for learning the student network are acquired from high-resolution images through bicubic interpolation. Note that these artificially synthesized low-resolution images are also used for testing of the proposed method. The values obtained by sampling from 1.1 to 4.0 at intervals of 0.1 are used as the scale factor *s*, and in this case, the smallest resolution is 9×15. Three-fold cross-validation and leave-one-out cross-validation are performed for UTMV and MPII, respectively. 50 UTMV subjects are divided into three groups, {0–16}, {17–33}, and {34–49}, based on the subject ID to conduct the experiment.

### 4.2. Implementation Details

The teacher and student networks are constructed by modifying the FC layer of ResNet-18 [44]. Their hyperparameters, such as kernel size and stride of the convolutional layer, are all the same as in ResNet-18. As ResNet is a model for classifying input images into 1000 classes, the existing FC layer is removed and then two FC layers and a ReLU activation function are added for gaze estimation. The size of the two added FC layers is 128 and 3. Our framework uses four residual modules and four feature adaptation modules (i.e., N=4).

The values of {kernel size, stride, padding} of the four convolutional layers in the feature adaptation module are {3, 1, 1}. Except in the last layer, the convolutional layer is followed by batch normalization and ReLU activation functions. The activation function following the last convolutional layer is a sigmoid function, as shown in Figure 3a. The number of output channels depends on the number of input channels. Assuming that the number of input channels is *C*, the number of output channels is reduced to C/4, C/16, C/16, 1. The size of the FC layers in the scale-conditioned layer is 64 and 4, and the size of the expert is C×C×3×3.

In the student network, the parameters of all layers, except the feature adaptation module, are initialized to be the same as the parameters of the pre-trained teacher network. Adam [45] is used with a momentum of 0.9 for training the teacher and student networks. In all experiments, the learning rate is kept constant at 5 × 10^−3^, and the mini-batch size is 256. λ in Equation (Equation 3) is experimentally set to 10.

### 4.3. Ablation Study

Our framework uses knowledge distillation and feature adaptation together. For the ablation study, the feature adaptation module is removed from the student network, and the network is learned using only $L_{gaze}^{S}$ without distillation loss. The trained model is used as a baseline for comparison with the proposed method. That is, **Baseline** represents a general backbone-based model trained using only the gaze loss $L_{gaze}^{S}$ from input images of various sizes. Figure 4 shows the results of ablation experiments, which proves that all components used in the proposed method (i.e., feature adaptation (**FA**) and knowledge distillation (**KD**)) contribute to the performance improvement. **Baseline + FA** and **Baseline + KD** represent models in which the feature adaptation module and distillation loss $L_{KD}^{S}$ are added to the baseline, respectively. **Baseline + FA + KD** is the proposed method, which uses knowledge distillation and feature adaptation at the same time. **HR image** represents the gaze estimation result of the teacher network on the high-resolution image, which can be considered the most positive performance expectation that the proposed method can perform.

According to the experimental results, **Baseline + FA** achieves slightly improved results compared to **Baseline**, which shows that the capacity increase in the model due to the addition of the feature adaptation module helps the model adapt to input images of various sizes. However, this result corresponds to lower performance compared with **Baseline + KD**. Such an outcome reveals that the teacher’s information transferred through knowledge distillation is more effective for gaze estimation than the increase in the model’s capacity. Finally, **Baseline + FA + KD**, which is the proposed method, shows the best performance for all scale factors by using information extracted from high-resolution images and increasing the model’s capacity. This result justifies that the proposed components (**FA** and **KD**) improve the performance of our method.

The UTMV dataset is acquired in a laboratory environment with eight fixed head poses. However, the MPII dataset is acquired in an unconstrained environment, so the head pose variation is more diverse. The roll rotation of the head pose is removed through preprocessing in both datasets. Still, the processed images in the MPII dataset contain more pitch and yaw rotation variations than those in the UTMV dataset. Therefore, the MPII dataset is more complex and challenging than the UTMV dataset. Nevertheless, the performance improvement of gaze estimation by the proposed method is higher in the MPII dataset than in the UTMV dataset. In particular, when the scale factor is smaller than 2.5, the proposed method shows comparable performance with the **HR image** result. Therefore, performance improvement by the proposed method (**Baseline + FA + KD**) over **Baseline** is valid for input images containing various head poses.

For statistical tests, the paired t-test is performed using the mean gaze errors of the proposed method (**Baseline + FA + KD**) and the **Baseline** over 15 leave-one-person-out cross-validation folds of the MPII dataset. Figure 5 visualizes the calculated *p*-values. For all scale factors except s=1.1, the *p*-value is less than 0.05. As the scale factor increases, the *p*-value decreases. This result shows that the performance gain of the proposed method is statistically significant, especially for high scale factors.

### 4.4. Comparison to Baselines

Gaze estimation for an arbitrary-sized low-resolution input image can be performed through two other methods, as shown in Figure 1b,c. The first method (i.e., **Arbitrary SR baseline**) applies SR to a low-resolution image of any size and uses the resultant image for gaze estimation. Among the SR methods for low-resolution images of arbitrary size, Meta-RDN [19] is used in our experiments. In the second method (i.e., **Multiple gaze baseline**), different gaze estimation models are trained according to various resolutions of input images. To this end, various teacher networks corresponding to various resolutions are trained. For comparison with the above two baselines, 30 scale factors (1.1, 1.2, *…*, 4.0) are used; thus, a total of 30 networks are trained for the **Multiple gaze baseline**. In the case of the **Arbitrary SR baseline**, the FLOPs in Meta-RDN vary according to the scale factor, so the average value for 30 scale factors is calculated.

Table 1 shows the results of comparing our method with the above two baselines. Params and FLOPs represent the numbers of learnable parameters (in mega) and floating point operations (in giga) of the model, respectively. The last six columns represent gaze estimation errors according to integer scale factors (i.e., 2, 3, and 4) on UTMV and MPII datasets. According to Table 1, the **Arbitrary SR baseline** shows a gaze estimation error of approximately 0.2 degrees lower than that of our framework. However, this baseline has about three times as many parameters and about 100 times more FLOPs than our framework. The **Multiple gaze baseline** has the highest number of parameters but the smallest number of FLOPs. It is because there are 30 different models for 30 scale factors, and the gaze vector is estimated through just one model in the inference time. The proposed method has almost the same FLOPs as the **Multiple gaze baseline** but shows fewer parameters and gaze estimation errors. Through this experiment, it is demonstrated that the proposed method can perform efficient gaze estimation in terms of memory and inference time for arbitrary-sized low-resolution images.

### 4.5. Generalization of the Proposed Method

ResNet [44] and VGG [46] are used as feature extractors (i.e., backbones) in various gaze estimation studies [14,16,25,34,35,47,48]. In the proposed method, ResNet-18 is used as the backbone. It is important to determine whether the ideas of feature adaptation and knowledge distillation proposed in this study can be applied to other backbones with various complexity. Therefore, experiments are performed by changing the existing ResNet-18-based teacher and student structure to investigate the generalization performance of the proposed method to other models. The teacher network based on ResNet-18 includes four residual modules. The number of building blocks [44] that compose each residual module is [2,2,2,2]. The number of these building blocks is changed to create simple and complex models compared with our original teacher network. The models used in the experiment are ResNet-10 and ResNet-34, which are composed of building blocks of [1,1,1,1] and [3,4,6,3], respectively. In the case of VGG, VGG-13, VGG-16, and VGG-19 with batch normalization are used. Experiments are conducted using the MPII dataset acquired in an unconstrained environment.

Figure 6 shows the experimental results for the two backbones. The results indicate that the proposed method improves the gaze estimation performance for arbitrary-sized low-resolution images regardless of the structure or complexity of the model. In the case of the ResNet backbone, increasing the model complexity improves the gaze estimation performance of the baseline and the proposed method. In the case of the VGG backbone, increasing model complexity does not result in performance improvement. VGG has relatively many parameters compared with ResNet, which causes overfitting. As a result, the simplest VGG-13-based model (**VGG13 + FA + KD**) shows the best performance for most scales.

### 4.6. Model Compression

Knowledge distillation was originally proposed to make simple models perform similarly to complex models with high performance [22]. Therefore, most studies using knowledge distillation focus on reducing the model size. In our study, the distillation loss only transfers knowledge from the teacher network to the student network. In this section, the model compression aspect of knowledge distillation is further investigated. In Section 4.5, the performance of the ResNet backbone increases as the model becomes more complex. However, the performance of the VGG backbone does not increase due to overfitting, so an experiment on model compression is conducted using the ResNet backbone.

Figure 7 shows the experimental results of model compression using ResNet. ResNet-34 and ResNet-18 are used for the teacher network in Figure 7a,b, respectively. In both cases, the student network is based on ResNet-10. According to Figure 7a, ResNet34→ResNet34 outperforms ResNet10→ResNet10. This finding reveals that using a complex backbone improves gaze estimation performance. Using a lighter student network (ResNet34→ResNet10) slightly reduces the performance compared with ResNet34→ResNet34, but it still achieves better performance than ResNet10→ResNet10. Figure 7b also shows similar experimental results. Model compression is evidently effective in the proposed method.

### 4.7. Qualitative Results

Figure 8 shows the visualization of the gaze vector estimated through the proposed method. The top three and bottom three rows show the results for the UTMV and MPII datasets, respectively. Each column shows the results for a scale factor sampled at intervals of 0.5 from 1.0 to 4.0. A scale factor of 1.0 means that a high-resolution (36×60) image is used. The blue arrow indicates the ground-truth gaze vector projected on the image. The green and red arrows correspond to the results estimated through the proposed method and baseline, respectively. As shown in Figure 8, the gaze vector inferred by the proposed method points in a direction more similar to the ground-truth vector than the gaze vector from the baseline. Hence, the proposed method can perform a more accurate gaze estimation than the baseline from input images of various resolutions.

### 4.8. Quantitative Comparison with Gaze Estimation Methods from Fixed-Sized Images

In most existing appearance-based gaze estimation methods, it is assumed that an input eye or face image of a fixed size is given. Although the goal of our study is different from theirs, the performance of the proposed method on fixed-size input images is presented for completeness. Among the various backbones in Section 4.5, a VGG13-based model (**VGG13 + FA + KD**) is selected and used for gaze estimation from fixed-size input images of the MPII dataset. A quantitative comparison with the existing methods is given in Table 2. The proposed method achieves the best performance among the compared eye-image-based methods. These results show that the proposed method achieves competitive performance with other recent methods for fixed-size input images, even though our approach was learned for multiple scale factors for gaze estimation from arbitrary-sized low-resolution images.

### 4.9. Discussion

Our thorough experiments show that the two contributions of our study presented in Section 1 are achieved. First, as reviewed in Section 2, existing gaze estimation studies rarely address arbitrary-sized low-resolution images [6,24,25,26,27]. In our study, two baselines visualized in Figure 1 are implemented to address the problem and quantitatively compared with the proposed method in Section 4.4. Compared to the **Arbitrary SR Baseline**, which depends on an external SR network, and the **Multiple gaze baseline**, which depends on multiple gaze models, the proposed method consists of a single novel model, so it can efficiently perform gaze estimation. Second, the proposed method is based on a novel combination of knowledge distillation [22] and feature adaptation [19,20,23,43]. According to the experiments in Section 4.3, both components contribute substantially to the performance improvement of the proposed method for gaze estimation compared to the naive **Baseline**.

However, the method proposed in this study has the following limitations. First, as mentioned in Section 1, in the proposed method, the aspect ratio of the input image is assumed to be constant, which means that the size of the input image is not genuinely arbitrary. Since human eyes are considered to have a similar aspect ratio, the aspect ratio of the input images will not be much different. However, the proposed method requires input images of a fixed aspect ratio, which is one of the limitations of the proposed method. Next, as mentioned in Section 4.1, the MPII and UTMV datasets used for training and evaluating the proposed method consist of high-resolution images. Therefore, the low-resolution images for learning the proposed method are artificially synthesized through downsampling from the corresponding high-resolution images. These synthetic images are not guaranteed to have the same distribution as real low-resolution images. Therefore, another limitation of our study is that the proposed method is trained and evaluated on synthetic images rather than real ones.

## 5. Conclusions

In this study, a gaze estimation method (i.e., ArbGaze) from arbitrary-sized low-resolution images is proposed. The proposed method adopts the teacher–student framework and transfers the teacher model information obtained from high-resolution images to the student, a gaze estimation model for arbitrary-sized images, through knowledge distillation. The feature adaptation module helps the student network effectively learn the information transferred from the teacher network through CondConv parameterized with scale information. It is empirically proved that the proposed method is more efficient and effective than baseline models that can perform gaze estimation from arbitrary-sized low-resolution images. In addition, it is demonstrated that the proposed method can be easily generalized to other gaze estimation models through experiments on various backbones. Our future work is to extend the proposed approach to other computer vision tasks and apply ArbGaze to various application areas, such as human–computer interaction, human–robot interaction, virtual reality, augmented reality, and autonomous driving.

## Figures and Tables

**Figure 1 sensors-22-07427-f001:**
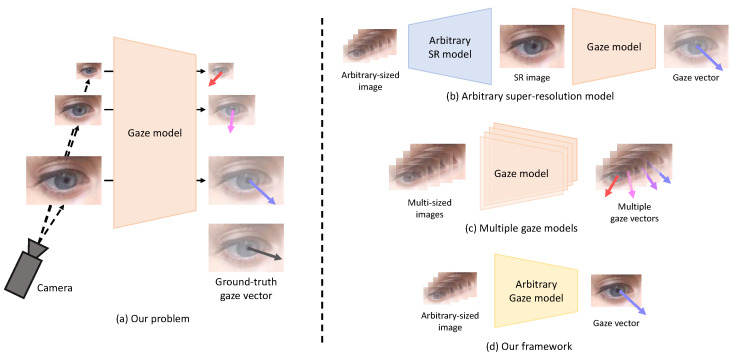
The problem to be addressed in this study (i.e., gaze estimation from arbitrary-sized low-resolution images) is shown in (**a**). Low-resolution images of various sizes can degrade the gaze estimation performance of the existing gaze estimation model. The following approaches can be used to solve this problem: (**b**) a method based on an arbitrary SR model, (**c**) a method based on several different gaze estimation models corresponding to various resolutions, and (**d**) the proposed method that is based on a single model.

**Figure 2 sensors-22-07427-f002:**
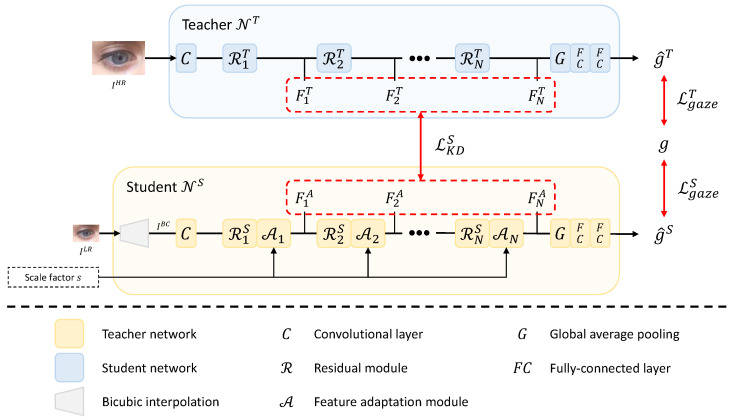
Overview of the proposed method.

**Figure 3 sensors-22-07427-f003:**
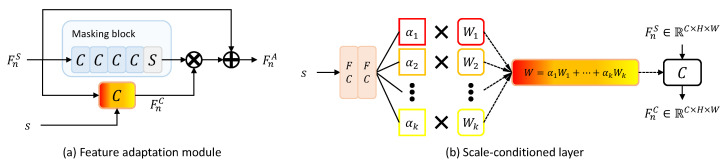
The feature adaptation module and the scale-conditioned layer are illustrated in (**a**,**b**), respectively. The weights of the convolutional kernel *W* are calculated by a linear combination of *K* routing weights and experts.

**Figure 4 sensors-22-07427-f004:**
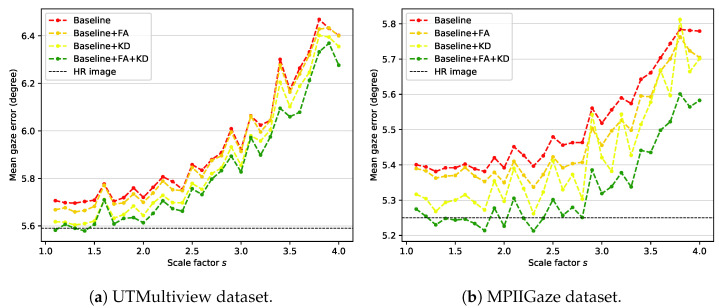
Mean gaze error over all scale factors on (**a**) UTMV dataset and (**b**) MPII dataset.

**Figure 5 sensors-22-07427-f005:**
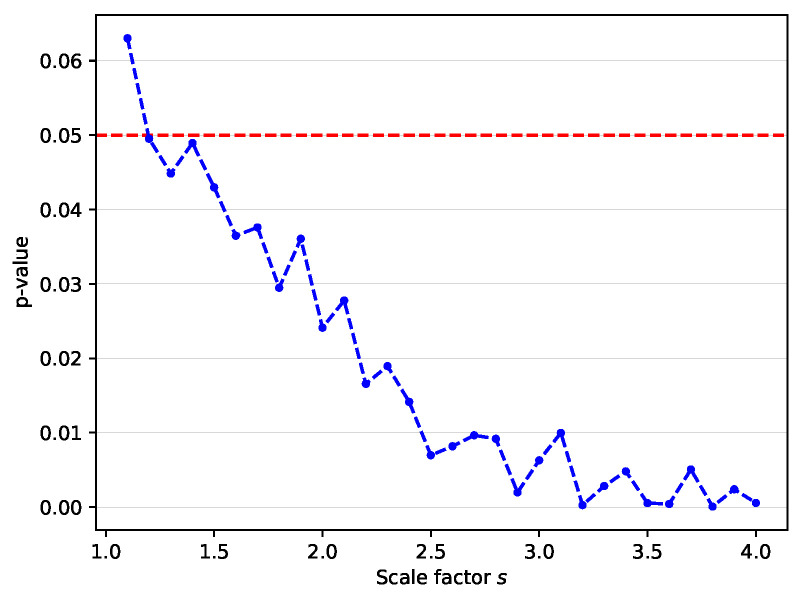
*p*-value over all scale factors on MPII dataset. This result demonstrates that the improved performance of the proposed method with respect to the baseline is significant at α=0.05.

**Figure 6 sensors-22-07427-f006:**
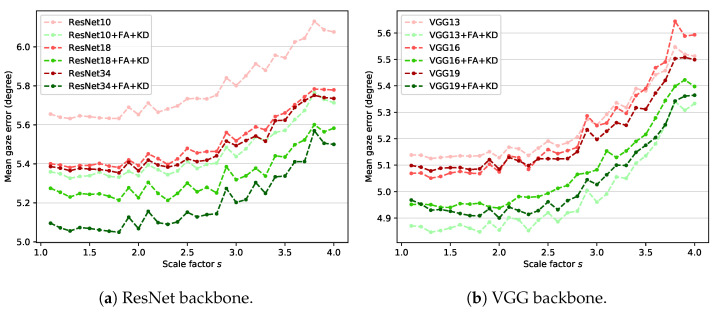
Mean gaze error on MPII dataset with different backbones: (**a**) ResNet and (**b**) VGG.

**Figure 7 sensors-22-07427-f007:**
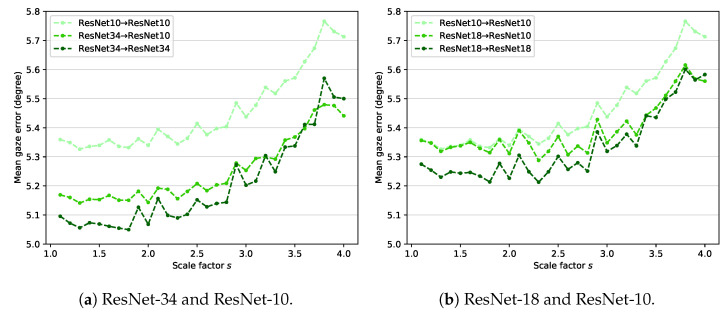
Results of model compression. (**a**) ResNet-34 and ResNet-10 are used for the teacher–student framework. (**b**) ResNet-18 and ResNet-10 are used for the teacher–student framework.

**Figure 8 sensors-22-07427-f008:**
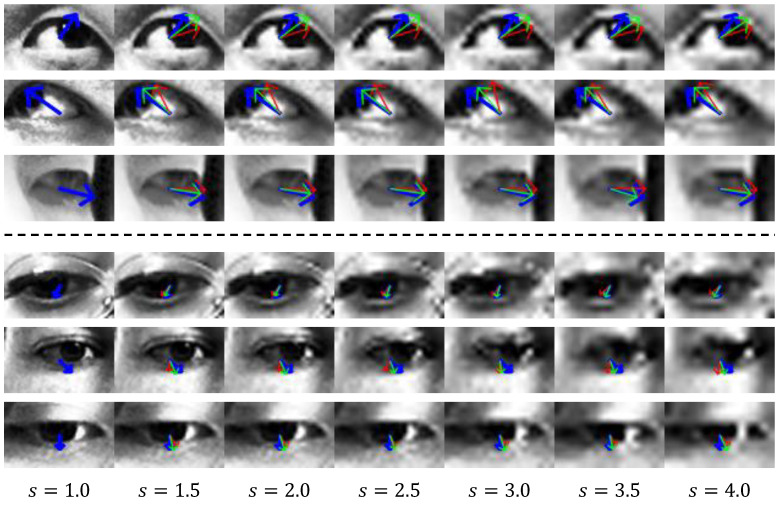
Gaze vector visualization. The top and bottom three rows correspond to the UTMV and MPII datasets, respectively. *s* denotes the scale factor. The blue arrow indicates the ground-truth gaze vector. The green and red arrows indicate estimated gaze vectors by the proposed and baseline methods, respectively.

**Table 1 sensors-22-07427-t001:** Quantitative comparison to other baselines for gaze estimation from arbitrary-sized low-resolution images. The best results are shown in **bold**, and the second best results are underlined.

Method	Params (M) ↓	FLOPs (G) ↓	UTMV (Degree) ↓			MPII (Degree) ↓		
			*s* = 2	*s* = 3	*s* = 4	*s* = 2	*s* = 3	*s* = 4
Arbitrary SR baseline	33.36	23.51	5.60	**5.73**	**5.97**	**5.21**	**5.26**	**5.37**
Multiple gaze baseline	337.09	**0.20**	**5.47**	5.90	6.45	5.32	5.60	5.82
ArbGaze (ours)	**12.08**	0.22	5.61	5.83	6.28	5.23	5.32	5.58

**Table 2 sensors-22-07427-t002:** Quantitative comparison to other existing gaze estimation methods from fixed-sized images on MPII dataset. The best results are shown in **bold**, and the second best results are underlined.

Method	Input	Gaze Error (Degree) ↓
RF [10]	eye image	7.99
Mnist [17]	eye image	6.30
GazeNet [48]	eye image	5.83
ARE-Net [49]	eye image	5.02
Proposed (VGG13 + FA + KD)	eye image	4.88
Full Face [7]	face image	4.90
RT-Gaze [14]	face image	**4.30**
FAR-Net [50]	face image	**4.30**

## Data Availability

Not applicable.

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
