# Peer review of "ArbGaze: Gaze Estimation from Arbitrary-Sized Low-Resolution Images"

_sensors, 2022, doi:10.3390/s22197427_

Round 1
Reviewer 1 Report
This paper proposes the use of knowledge distillation and feature adaption to allow the use of arbitrary resolution images to estimate gaze direction. This is worthwhile research with intriguing results; however some limitations of the work need to be more transparently addressed and discussed.
Detailed comments:
1. Line (Ln) 45-51: The teacher-student analogy in not explained well upon introduction
2. Ln 95-104: More details on the related work, issues and solutions would be appreciated, particularly how they are relevant/related to your problem and solution
3. Ln 133-140: Why aren’t these methods good enough? Is your method clearly superior? A comparison of results may be in order.
4. Ln 151-152: Up until now, you said your method would work with any arbitrary resolution, but now you introduce the limitation that the aspect ratios must be the same. This reduces the general applicability and means your input resolution can no longer be truly arbitrary. This should be addressed up front and as a limitation of the method.
5. Ln 205: Before jumping into the results, a subsection of your method section could explain your intended method of testing, and its rational. This would help prep the reader for the upcoming tests, and interpreting the results.
6. Ln 220-224: given your source dataset, you do not test on true arbitrary resolution images, but (artificial) estimations of them. This should be noted.
7. Ln 250: Ablation studies allow us to see the effect of a method/algorithm comparted to baseline (nothing). This alone does not prove the method is worthwhile in the real world. A small increase may not be worthwhile, especially if better methods exist. So be careful not to over-sell the significance of the ablation study.
8. Table 1: Did you determine the statistical significance of these results?
9. Figure 7: the MPII Gaze results are too small to interpret well; a larger image would be helpful
10. Line 352: A section discussing the limitations of the proposed method is warranted.
Reviewer 2 Report
Well written paper, few suggestions for authors
-Introduce the motivation of the study in the light of the literature review
-Please introduce the Application section where authors can celebrate the application of this project
-Include research questions and hypothesis
-Baseline results should be compared with statistical testing with a specific confidence
-Please use discussion sections before the conclusion and discuss your results in light of the literature review.
Reviewer 3 Report
The organization of the paper is good. However some improvements are required:
1) The authors mentioned 2 contributions of their works. They need to write clearly in discussion sections or at the end of experimental results section that they have achieved 2 contributions throughout the different dimension of experiments.
2) ResNet and VGG are used for feature extractions in this paper. They need to justify why they have used these two for feature extractions. What is the reason behind changing the existing ResNet-18-based teacher and student structure.
3) Other than the ResNet and VGG, the authors should compare their model with others network classifiers.
4) Authors should compare their results with some other existing recent works. The result should be presented in tabular form for better understanding,
5) Write the paper in passive voice. Don’t write : I, we and so on.
6) Please read and cite some quality works in this area : “Eye gaze estimation: A survey on deep learning-based approaches (2022)”, “A Human-Robot Interaction System Calculating Visual Focus of Human’s Attention Level (2021)”, “When i look into your eyes: A survey on computer vision contributions for human gaze estimation and tracking (2020)”, “Appearance-based gaze estimation with deep learning: A review and benchmark (2021)”,.
Round 2
Reviewer 3 Report
All comments are addressed successfully